# Selection and Validation of Suitable Reference Genes for RT-qPCR Analysis in the Rare Aquatic Firefly *Aquatica leii* (Coleoptera: Lampyridae)

**DOI:** 10.3390/insects12040359

**Published:** 2021-04-16

**Authors:** Xinhua Fu, Victor Benno Meyer-Rochow

**Affiliations:** 1College of Plant Science and Technology, Huazhong Agricultural University, Wuhan 430070, China; 2Firefly Conservation Research Centre, Wuhan 430070, China; 3Department of Ecology and Genetics, Oulu University, SF-90140 Oulu, Finland; meyrow@gmail.com; 4Agricultural Science and Technology Research Institute, Andong National University, Andong 36729, Korea

**Keywords:** reference gene, selection, validation, RT-qPCR, development, RNAi, firefly, *Aquatica leii*

## Abstract

**Simple Summary:**

The purpose of this paper was to determine which genes in *Aquatica leii*, an endemic Chinese firefly, are suitable for use as reference genes in future RT-qPCR studies. This is the first example of an effective knockdown in fireflies and an important methodological advance for continuing research in *A. leii*. The results of this study will help improve accuracy and reliability to normalize RT-qPCR data in *A. leii* for further molecular analysis. Since luciferase is very abundant in the adult light organ of fireflies generally, this is an encouraging result and of interest not just for *A. leii* researchers but other researchers who plan to perform RNAi or RT-qPCR in fireflies, or other nonmodel insects, as well.

**Abstract:**

*Aquatica leii* Fu and Ballantyne is a species of rare aquatic firefly and endemic in China. It is considered good material to study the molecular mechanism of sexual flash communication systems. To improve conservation and behavioral research strategies, large-scale genetic studies involving gene-expression analysis are required and reverse transcription-quantitative polymerase chain reaction (RT-qPCR) is the most commonly used method. However, there have been very few reports on appropriate reference genes in any species of firefly. Here, we evaluated eight widely utilized reference genes including *18S*, *Actin*, *Reep5*, *Odc1*, *Tub*, *Gapdh*, *Ef1a* and *S27Ae* for their expression stabilities in *A. leii* under three different conditions, i.e., life stage, tissue and dsRNA injection. Based on the gene stability ranking calculated by RefFinder, which integrates four algorithms (geNorm, delta Ct method, NormFinder, and BestKeeper), we recommend *S27Ae* and *Reep5* as the most appropriate reference genes for molecular studies in different life stages; *Ef1a* and *Odc1* for different tissues; *Tub* and *Odc1* for RNAi studies. The most appropriate reference genes in all treatments are *S27Ae* and *Tub***.** The results of this study will help improve accuracy and reliability to normalize RT-qPCR data in *A. leii* for further molecular analysis.

## 1. Introduction

*Aquatica leii* Fu and Ballantyne 2006 (Coleoptera: Lampyridae) is an important and rare aquatic endemic firefly in China. The species is very sensitive to water quality and pollution and is considered endangered, with populations known to be in decline [1]. The larvae of *A. leii* live in standing and clean water and feed on freshwater snails, a feature potentially useful in the biological control of various parasites that use freshwater snails as intermediate hosts [2]. Light organ sexual dimorphism is distinct, with two ventral segments (six and seven) in males but only one ventral segment (six) in females. Adults use species-specific flash signals during courtship. Males transfer a nuptial gift, i.e., the spermatophore to the female during mating [1]. To study the molecular mechanism of flash communication behavior of *A. leii*, reference genes must first be selected. The *Fruitless (Fru)* gene probably acts as a transcriptional regulator. As part of the somatic sex determination hierarchy, the sex determination genes *transformer* (*Tra*) and *transformer*-2 (*Tra*-2) switch *Fru* splicing from the male-specific pattern to the female-specific pattern through activation of the female-specific *Fru* 5’-splice site. The gene *Fru* is vital for the development of male and female characteristics. It controls the development of the male-specific abdominal muscle and plays a role in male courtship behavior and sexual orientation. Besides, it also enhances male-specific expression of takeout (a protein related to a superfamily of factors that bind small lipophilic molecules) in brain-associated fat body functions [3].

No studies to date have attempted to reveal the underlying molecular regulatory mechanisms of *Fru* in firefly courtship behavior and flash communication. Thus, an understanding of the gene expression patterns may offer clues to complex regulatory networks and help us identify genes relevant to novel biological processes such as the sex-determination pathway and male courtship behavior in fireflies generally. Toward this end, we screened and evaluated candidate reference genes using quantitative reverse-transcription PCR (qRT-PCR) to measure the expression across different samples.

Real-time quantitative RT-PCR (RT-qPCR) is the most commonly used technology for accurately detecting gene expression, and it has been widely applied in molecular biology [4,5]. Compared with other traditional molecular techniques, RT-qPCR has the advantages of higher sensitivity, better reproducibility and specificity. It has become the standard for gene expression quantification with the character of high-throughput [6,7]. However, the results of RT-qPCR vary due to differences in initial sample size, template RNA integrity, mRNA recovery, reverse transcription efficiency, and primer design. To obtain accurate and reliable gene expression results, RT-qPCR data must be normalized with appropriate reference genes, the expression of which should stabilize during treatment. The most frequently used reference genes might not be stably expressed under different experimental conditions, causing a high risk of result misinterpretation. Therefore, it is necessary to perform systematic selection and validation of the reference genes to maximize the accuracy of PCR analysis and the reliability of the gene expression data [5,8]. Our study investigated the *Fru* gene expression of different tissues and stages in *A. leii*. Furthermore, the luciferase gene *Luc* expression level was also investigated after RNAi interference. Screening reference genes is considered a necessary step in relation to different tissues and stages.

In this study, we report quantitative analyses of the expression of eight candidate reference genes in various tissues, stages and treatments of *A. leii*. The eight genes evaluated were *β-Actin* (*Actin*), *18S* ribosomal RNA (*18S rRNA*), ornithine decarboxylase1(*Odc1*), glyceraldehyde-3-phosphate dehydrogenase (*Gapdh*), elongation factor 1-alpha (*EF1-α*), Receptor expression-enhancing protein 5 (*Reep5*), Ubiquitin-40S ribosomal protein S27Ae (*S27Ae*) and *β-tubulin* (*Tub*), all of which have been widely used as reference genes in different organisms because they are considered to have a uniform expression. Those eight reference genes were chosen because they have been used to investigate different insects’ tissues or stages during RT-qPCR experiments [9,10]. This study provides the first reliable reference for the selection of reference genes for firefly gene expression studies.

## 2. Materials and Methods

### 2.1. Insects

Larvae, pupae and adults were obtained from aquatic firefly *A. leii* breeding lab (a lab established solely to breed *A. leii* the original firefly population was collected from Hangzhou city) and kept in the laboratory at 25 ± 1 °C for 24 h under 70 ± 5% humidity and a 14:10-h light/dark (L:D) photoperiod.

### 2.2. Sample Preparation

#### 2.2.1. Developmental Stage

Five different stages were examined. They included 3rd and 5th instar larvae, 1d-pupae, 3d-pupae and 5d-pupae.

#### 2.2.2. Tissue

Six different tissues of adult individuals were examined. They included the head-thorax, light organ, fat body, testis, ovary and the remainder of the body.

#### 2.2.3. dsRNA Injection

For RNAi treatments, 1-d pupae of *A. leii* were microinjected with 750 nl dsRNA (2 μg/μL) against *Luc*, which is luciferase, a participant in the bioluminescence of fireflies. The control consisted of injections of *dsGfp* (dsRNA against green fluorescent protein). The dsRNAs were synthesized using a TranscriptAid T7 High Yield Transcription Kit (Thermo Scientific, Waltham, MA, USA, #K0441) following the manufacturer’s instructions. The corresponding primers are shown in Table 1. On day one after emergence, three biological replicates were collected per treatment. Each replicate consisted of three individuals. All samples were frozen at −80 °C until RNA extraction was performed. The reasons why RNAi samples against *Luc* was chosen is that the *Luc* gene is a unique gene in the firefly *A. leii* and responsible for bioluminescence, easy to observe and stable. Besides, to explore the relationship between *Luc* and *Fru* was one of the objectives of this study.

### 2.3. RNA Extraction and cDNA Synthesis

Total RNA of the above samples was extracted using TRIzol Reagent (Invitrogen, 15596018). The RNA concentrations were determined on a Nano-Drop 2000 spectrophotometer (Thermo Scientific), and further checked by 1.5% agarose gel electrophoresis. Subsequently, 1 μg total RNA was reverse-transcribed with the RevertAid First Strand cDNA Synthesis Kit (Thermo Scientific, #K1621) and DNase I (Thermo Scientific, #EN0521).

### 2.4. Candidate Reference Genes and Primer Design

Eight commonly used reference genes were selected (Table A1). The full lengths of the above reference genes were amplified using primers (Table 1) designed and based on our recently sequenced *A. leii* transcriptome data (unpublished data), cloned into the pMD18-T vector (Takara, D101A), and the sequenced. The candidate reference genes were subsequently confirmed and submitted to GenBank (Table 1). The primers used for qRT-PCR were designed online (https://primer3.ut.ee/, accessed on: 18 April 2021) with the following criteria: GC content 50–60%, optimal Tm 60–62 °C, primer length 20–22 bp, and amplicon length 90–230 bp.

### 2.5. Quantitative Reverse Transcription-Polymerase Chain Reaction

qRT-PCR analysis was conducted in a CFX Connect Real-time PCR Detection System (Bio-Rad, Hercules, CA, USA). The reactions were prepared as follows: 2 μL of 20-fold diluted cDNA template, 10-μL 2 × Power SYBR^®^ Green PCR Master Mix (Applied Biosystems, 4367659), 0.5 μM of each gene-specific primer (Table 1), and ddH2O for the remaining volume. The PCR program consisted of an initial denaturation at 95 °C for 3 min, followed by 40 repeated cycles, each consisting of 95 °C for 10 s, 55 °C for 20 s, 72 °C for 20 s, 75 °C for 5 s, and plate read. The melting curve covered 65 °C to 95 °C, with increments 0.5 °C for 5 s, and then plate read.

### 2.6. Stability Evaluation of Candidate Reference Genes

The ΔCt method, and the programs geNorm, NormFinder, and BestKeeper were used to evaluate the stability of all nine potential reference genes. The Ct value is the number of amplification cycles that are elapsed when the fluorescence signal of the amplified product reaches the set threshold during PCR amplification. Data analyses were analyzed independently for each adult tissue of Ct values. The stability of the reference gene means that the expression levels should be approximate and no significant differences were observed under various types of tissues and various experimental conditions. The stability of the 8 candidate reference genes was evaluated using geNorm, NormFinder, and Best-Keeper as well as the delta Ct method in Microsoft Excel. The geNorm algorithm determines an expression stability value (M) for each gene and then compares the pairwise variation (V) in this candidate reference gene with that in other tested candidate reference genes. Candidate reference genes with lower M-values have more stable expression [5]. NormFinder uses a model-based method to estimate the variation in expression of candidate reference genes, assigning a stability value to each candidate reference gene, whereby candidate reference genes with lower values are identifiable as more stable [11]. BestKeeper calculates the standard deviation (SD) and stability value (SV) of candidate reference genes based on raw data (CT values), and those with low index scores are considered to be highly stable [12]. The delta Ct method calculates the mean SD by pairwise comparisons; a lower SD being indicative of a more stable gene [13]. Finally, RefFinder, a web-based comprehensive algorithm used to evaluate and screen candidate reference genes, integrates four computational programs (geNorm, NormFinder, BestKeeper, and delta Ct) to rank candidate reference genes. Based on the rankings from each program, it assigns an appropriate weight to an individual gene and calculates the geometric mean of their weights for the overall final ranking [14].

### 2.7. Validation of a Selected Reference Gene

*Fru* was selected to evaluate the reference gene. A full length of the *Fru* gene was amplified using primers (Table 1), with a design based on our recently sequenced *A. leii* transcriptome data (unpublished data), and then cloned into the pMD18-T vector (Takara, D101A), sequenced, confirmed and submitted to GenBank (Table 1). The gene expression was measured in various tissues and normalized by the optimal reference genes (*Tub*, *S27Ae*) and the least stable reference genes (*Actin*, *18S*). The qRT-PCR data were calculated using the 2^−ΔΔCt^ method [14]. Prior to analysis, the normality of all variables was tested using the Kolmogorov–Smirnov test and the homogeneity of group variances was assessed using Levene’s test. Finally, a statistical comparison was performed using IBM SPSS Statistics 24 and one-way analysis of variance followed by Tukey’s HSD Multiple Comparison.

## 3. Results

### 3.1. PCR Amplification of Candidate Reference Genes

Based on the results of the melting curves, a single peak but no signal in the negative controls for each reaction was obtained (Figure A1). This suggested that each gene was specifically amplified. Candidate reference gene sequences were amplified correctly by gel electrophoresis, and all the fragments were cloned and sequenced.

### 3.2. Expression Profiling of Candidate Reference Genes in A. leii

To identify stable reference genes, expression of the eight candidate reference genes across all samples was detected by qRT-PCR. The variations in candidate reference gene mRNA were revealed by the spectrum of Cp values across all samples. Theoretically, the candidate reference gene with the least amount of variation is the most stable one. In order to determine the dispersion of Cq values of the selected candidate reference genes under different physiological conditions, a boxplot comparison of the mean Cq values of all investigated candidate reference genes is shown in Figure 1. *18S* was the most abundant (the lowest Cq value) reference gene. In the tissue experiment, *Reep5* was the least abundant (the highest Cq value) reference gene and the highest variation in expression occurred with *Actin*. With regards to the developmental stage and the dsRNA injection experiment, *Reep5* was the least abundant and the reference gene with the highest variation in expression was *Tub*. In the total body experiment, *Reep5* was also the least abundant and the gene with the highest variation in expression was *Actin*. These results indicate that no candidate reference gene was consistently expressed across the different tissues, experimental treatments, or species. Therefore, identifying and targeting the most appropriate reference gene is necessary in order to understand normalizing gene expressions in a particular experimental system.

### 3.3. Expression Stability and Ranking of Candidate Reference Genes 

To identify the most appropriate reference gene(s) for the three experimental conditions (including different life stages, tissues and dsRNA treatments), the expression stabilities were analyzed by the ΔCt method, BestKeeper, NormFinder, and geNorm. RefFinder was the used to calculate an overall stability ranking. For the tissue-specific experiment, analyses using the ΔCt method showed that *Ef1a* and *Tub* were the most stable reference genes. BestKeeper showed that *18S* and *Odc1* were the most stable reference genes. The estimation by NormFinder suggested that *Odc1* and *Tub* were the most stable genes (Table 2). GeNorm gave *Ef1a* and *S27Ae* as the result of the most stable reference genes. Based on all four statistical algorithms we conclude that *Ef1a* and *Odc1* were the most stable reference genes. The stability of genes from most to least stable ranked by RefFinder was *Ef1a > Odc1 > Tub > S27Ae > 18S > Gapdh> Reep5 >Actin* (Figure 2). Thus, the best combination of reference genes for tissue samples of *A. leii* was *Ef1a* and *Odc1*.

For the different developmental stage experiment, the ΔCt method identified *S27Ae* and *Reep5* as the most suitable reference genes; the BestKeeper method showed *18S* and *Odc1* were the most stable reference genes while Normfinder came up with *S27Ae* and *Reep5* as the most stable reference genes and GeNorm identified *S27Ae* and *Tub* as the most suitable reference genes (Table 2). The overall ranking calculated by RefFinder was as follows: *S27Ae > Reep5 > Odc1 > Gapdh > 18S > Tub > Ef1a > Actin* (Figure 2). Thus, the best combination of reference genes in connection with different developmental stage in *A. leii* was *S27Ae* and *Reep5*.

For dsRNA injection experiments, based on the Normfinder and GeNorm methods, *Tub* and *Odc1* were the most stable reference genes; the ΔCt method identified *Tub* and *Ef1a* as the most suitable reference genes and the best reference genes from BestKeeper were *18S* and *S27Ae* (Table 2). The RefFinder ranking from highest to lowest stability was *Odc1 > Tub> Ef1a > S27Ae > 18S > Reep5 > Gapdh > Actin* (Figure 2). Thus, the best combination of reference genes for dsRNA injection experiments of *A. leii* were *Odc1* and *Tub*.

We therefore recommend *S27Ae* and *Tub* as the most appropriate reference genes for all treatments.

### 3.4. Validation of the Recommended Candidate Reference Genes 

To confirm whether normalization with the two most stable (*Tub* and *S27Ae*) or two least stable reference genes (*Actin* and *18S*) altered the qRT-PCR-measured expression of the genes of interest, the expression patterns of *Fru* in various tissues and different developmental stages were examined (Figure 3). We observed nearly nonluminescent adult phenotypes (Figure A2) and a significant gene knockdown in the firefly adults injected with dsRNA targeting *Luc*, compared with the control treatment injected with *Gfp* dsRNA (Figure A3). Therefore, we created a platform (Nanoliter2010/2T coupled with MICRO 2T SMARTouch; needle #504949, World Precision Instruments, ID = 0.530 mm ± 25 μm, OD 1.14 mm) of RNAi to fireflies after we had a nonluminescent firefly adult. In the ds RNA injection experiment, the expression patterns of *Fru* in *A. leii* were consistent regardless of using the two most or two least stable reference genes, which indicated that interference with the luciferase gene *Luc* had no influence on the *Fru* gene (Figure 3).

## 4. Discussion

Fireflies (Coleoptera: Lampyridae) are the most common representatives of bioluminescence [15,16,17]. The fascinating flash behavior in fireflies has recently been attracting more and more attention from scientists as diverse as phylogeneticists, molecular biologists, physiologists, ethologists and even psychologists. However, owing to increasing urbanization and pollution [18], the populations of many species of fireflies have been declining rapidly [15]. Until now, sparse research on the molecular mechanism of the development of light organs, flash behavior and conservation strategies has been conducted. With the advent of high-throughput sequencing technologies, increasing genetic information has been obtained from fireflies [19,20,21]. To further investigate the biological function of any particular gene within *A. leii*, the quantification of gene expression is essential, and requires reference genes with high stability. However, to date, no optimal reference gene (or combination of reference genes) has been identified and validated in any species of firefly. Normalizing expression data using unproven reference genes could lead to inaccurate data interpretation. Here, we identified the optimal reference genes for normalizing the gene expression data in different life stages and under different experimental conditions, including different developmental periods, tissues and dsRNA treatments.

We chose eight commonly used reference genes as candidate genes to evaluate their stability in different experimental conditions, tissues or life stages. In order to avoid errors in analysis caused by selecting coregulated transcripts, the candidate genes from different functional groups were selected according to four statistical models (ΔCt method, BestKeeper, NormFinder, and geNorm). The four analysis programs identified our different rankings (Table 2). The fluctuation of the ranking orders from the four analysis programs employed makes it difficult for researchers to choose optimal reference genes. Therefore, we used RefFinder, which integrates the abovementioned algorithms to rank the overall stability of candidate genes.

Our research also demonstrated that the choice of reference genes can affect experimental conclusions. Misinformed selection may lead to erroneous results, which will eventually result in the wrong targeted genes’ expression patterns. In our study, *Fru* exhibited inconsistent expression patterns in various tissues when normalized with stable or inappropriate reference genes (Figure 3). Our study shows that the use of inappropriate reference genes can statistically affect transcript quantification results and lead to misinterpretations and although Yang et. al. [22] have also recently investigated reference genes in *A. leii*, they did not select reference genes in the RNAi experiments as we did. Our data must therefore to be seen as an extension to the work of Yang et al. [22]. The findings of this study not only provide stable reference genes for the quantification of gene expression in *A. leii* but also lay the foundation for transcriptomics and functional gene research on fireflies generally.

## Figures and Tables

**Figure 1 insects-12-00359-f001:**
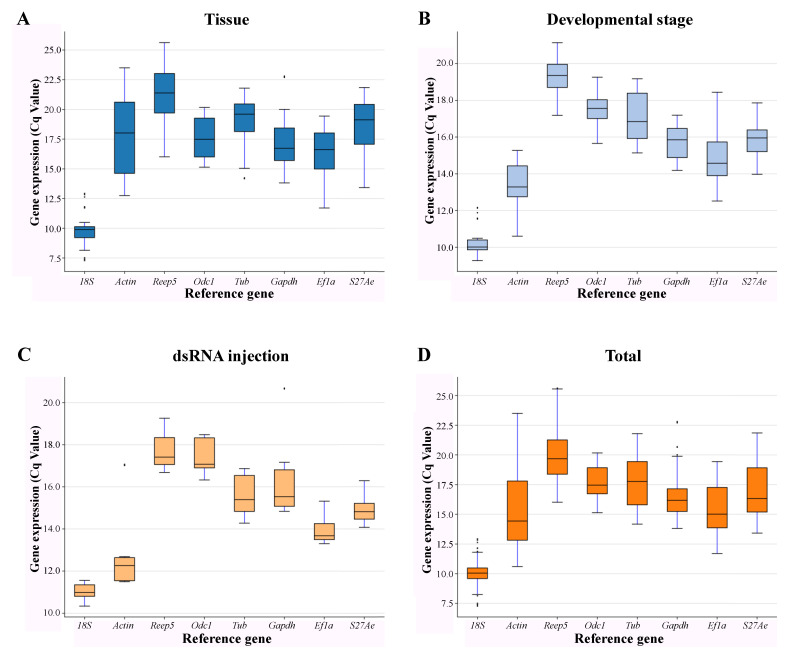
Expression profiles of candidate reference genes in different experimental conditions. Box and whisker plot chart showing the range of Cq values for each candidate reference gene under different treatments, tissue (**A**), developmental stages (**B**), dsRNA injection(**C**), and in all treatments (**D**). The upper and lower edges of the boxes indicate the 75th and 25th percentiles, respectively. Whiskers represent the minimum and maximum Cq values, the line within the box indicates the median. Small circles indicate the outliers.

**Figure 2 insects-12-00359-f002:**
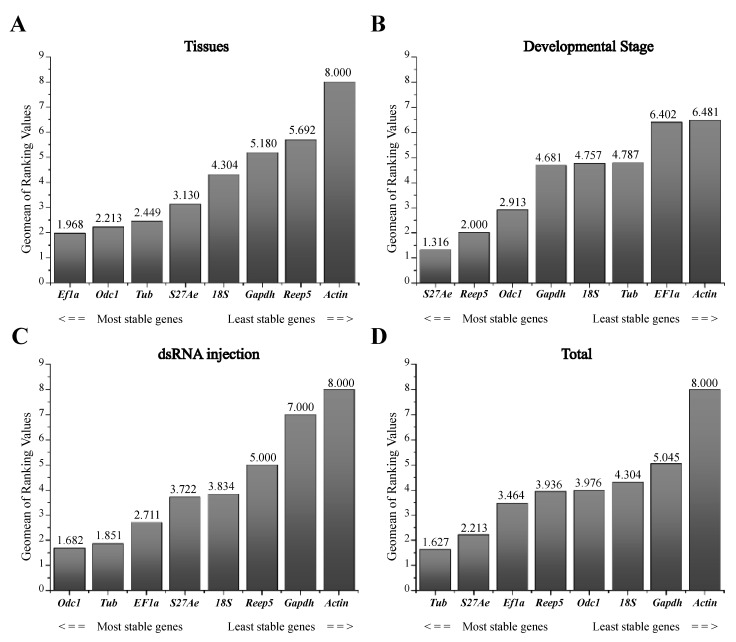
Stability of the reference genes in *A. leii* under different experimental conditions as determined by RefFinder.

**Figure 3 insects-12-00359-f003:**
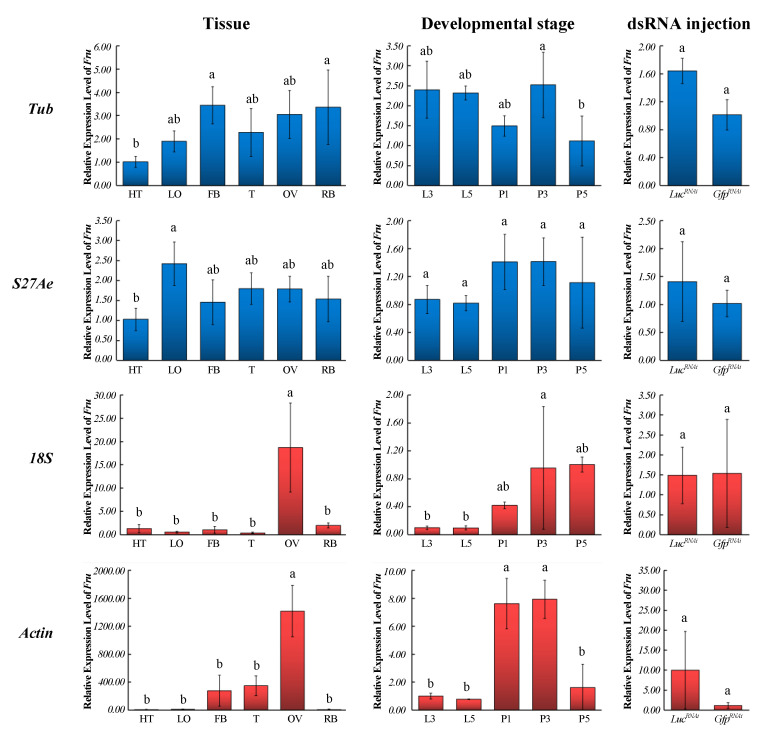
Relative expression levels of *Fru* in different tissues, developmental stages and treatment of RNAi interference against *Luc* gene. The relative mRNA expression levels of *Fru* were normalized to the most suited (*Tub* and *S27Ae*) and the least suited (*Actin* and *18S*) reference genes. Values are means ± SE. Different letters indicate statistically significant differences (*p* < 0.05, one-way analysis of variance followed by Tukey’s HSD Multiple Comparison). HT, head and thorax; LO, light organ; FB, fat body; T, testis; OV, ovary; RB, rest of body; L3, third-instar larvae; L5, fifth-instar larvae; P1, 1d- pupae; P3, 3d- pupae; P5, 5d- pupae.

**Table 1 insects-12-00359-t001:** Primers for the candidate reference genes and RNAi target genes used in the RT-qPCR analyses.

	Access Number	Primer Name	Primer Seq (5′-3′)	Primer TM (°C)	Length (bp)
*18S* (18S ribosomal RNA)	MT899427	18S F	TCTTAACCGAGTGTCCAGGC	56.43	158
		18S R	CATTACCTCTGTGCGTTCCA	54.91	
*Actin* (Actin, cytoplasmic 2)	MT899429	Actin F	GTATCTCACACCGTCCCCAT	55.83	143
		Actin R	CTTTCAGCGGTGGTTGTGAA	56.02	
*Reep5* (Receptor accessory protein 5)	MT899435	Reep5 F	ACCTACCGATTTCAATGGATCTC	58.04	119
		Reep5 R	GCTTTGCCGCTTCATTTTGG	59.21	
*Odc1* (Ornithine decarboxylase 1)	MT899431	Odc1 F	AGACGCTGAGTGGATTTTGC	58.84	101
		Odc1 R	CGTCCACATAATCCAGCACG	59.07	
*Tub* (Tubulin beta chain)	MT899434	Tub F	GTACGTTCGGGTCCATTTGG	58.92	114
		Tub R	GACCAATTCAGCACCTTCGG	59.2	
*Gapdh* (Glyceraldehyde-3-phosphate dehydrogenase)	MT899430	Gapdh F	ATCATTCCAGCAGCAACAGG	58.53	188
		Gapdh R	CCTTCGGCAGCTTCCTTTAC	58.91	
*Ef1a* (Elongation factor 1-alpha)	MT899428	Ef1a F	ATGGTTGTCGTCTTTGCACC	59.05	141
		Ef1a R	ACGACGCAATTCCTTAACGG	58.93	
*S27Ae* (Ubiquitin-40S ribosomal protein S27Ae)	MT899432	S27Ae F	TCCACCTGATCAACAACGTT	57.37	124
		*S27Ae* R	AGCACCACCTCGAAGTCTAA	58.37	
*Fru* (fruitless)	MT899433	Fru F	TCGCAAAACCTTCTTCCGAT	57.82	106
		Fru R	GCACTTCCGTTGTTTCGTCT	59.06	
*Luc*	MT990933	Luc F	GGAGATATTGGGTATTACGATG	ds RNA primer	378
		Luc R	CATCTTTGCATTTGGTTTCTTG		
*Gfp*	AAA27722.1	GFP F	CTACGGCGTGCAGTGCTTCAGC	ds RNA primer	350
		GFP F	AGTGGTCGGCGAGCTGCACGCTG		

**Table 2 insects-12-00359-t002:** Ranking of candidate reference genes according to different algorithms.

Candidate Genes	Δ*Ct* Method		BestKeeper		NormFinder		geNorm		Recommended Genes
	Average of STDEV	Ranking	Stability	Ranking	Stability	Ranking	Stability	Ranking	
Tissues									
*18S*	2.424	7	0.900	1	2.021	7	1.477	6	*Ef1a* & *Odc1*
*Actin*	3.457	8	2.915	8	3.292	8	1.972	7	
*Reep5*	1.780	5	2.110	7	1.136	6	1.012	4	
*Odc1*	1.559	3	1.549	2	0.325	1	0.886	3	
*Tub*	1.550	2	1.697	3	0.739	2	0.684	2	
*Gapdh*	1.850	6	1.748	4	1.008	5	1.193	5	
*Ef1a*	1.549	1	1.859	5	0.835	3	0.538	1	
*S27Ae*	1.604	4	2.109	6	0.994	4	0.538	1	
Developmental stages									
*18S*	1.648	8	0.507	1	1.437	8	1.264	7	*S27Ae* & *Reep5*
*Actin*	1.436	7	0.959	6	1.170	6	1.136	6	
*Reep5*	1.022	2	0.779	4	0.407	2	0.479	1	
*Odc1*	1.074	3	0.684	2	0.491	3	0.789	3	
*Tub*	1.270	5	1.218	7	0.975	5	0.665	2	
*Gapdh*	1.264	4	0.808	5	0.910	4	1.041	5	
*Ef1a*	1.423	6	1.240	8	1.191	7	0.896	4	
*S27Ae*	0.976	1	0.689	3	0.216	1	0.479	1	
dsRNA injection									
*18S*	1.344	6	0.323	1	1.046	6	0.744	5	*Odc1* & *Tub*
*Actin*	1.894	8	1.386	8	1.749	8	1.234	7	
*Reep5*	1.118	5	0.734	5	0.853	5	0.586	4	
*Odc1*	0.941	2	0.686	4	0.126	1	0.251	1	
*Tub*	0.929	1	0.742	6	0.126	2	0.251	1	
*Gapdh*	1.658	7	1.319	7	1.418	7	1.014	6	
*Ef1a*	0.948	3	0.513	2	0.454	3	0.492	2	
*S27Ae*	1.042	4	0.570	3	0.724	4	0.552	3	
Total									
*18S*	2.581	7	0.752	1	2.270	7	1.621	6	*Tub* & *S27Ae*
*Actin*	3.133	8	2.827	8	2.916	8	1.999	7	
*Reep5*	1.683	4	1.834	5	0.919	4	0.839	2	
*Odc1*	1.762	5	1.093	2	0.950	5	1.173	4	
*Tub*	1.646	1	1.926	7	0.864	1	0.781	1	
*Gapdh*	1.880	6	1.377	3	1.078	6	1.334	5	
*Ef1a*	1.660	3	1.700	4	0.902	3	0.922	3	
*S27Ae*	1.649	2	1.901	6	0.875	2	0.781	1	

## Data Availability

No additional data available.

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
