# Peer review of "Selection and Validation of Suitable Reference Genes for RT-qPCR Analysis in the Rare Aquatic Firefly Aquatica leii (Coleoptera: Lampyridae)"

_insects, 2021, doi:10.3390/insects12040359_

Round 1

Reviewer 1 Report

Here, Fu reports a set of RT-qPCR analyses for the purpose of determining which genes in Aquatica leii, an endemic Chinese firefly, are suitable for use as reference genes in future RT-qPCR studies. This is an important methodological advance for continuing research in Aquatica leii. Furthermore, the manuscript also presents proof-of-concept firefly dsRNA RNAi experiments. This manuscript will be interesting to other researchers who plan to perform RNAi or RT-qPCR in fireflies, or other non-model insects.

Overall, the technical portions of the RT-qPCR seem carefully performed. The conclusions are reasonable & supported by the data.

On the RNAi portions of the paper: One of the more interesting results of the paper is Figure S2/S3, showing successful knockdown of luciferase to a phenotypically impactful level. This is a first, to the reviewer’s knowledge, example of such an effective knockdown in fireflies. Luciferase is very abundant in the adult light organ, so this is an encouraging result.

The grammar and word choice, in many parts, is hard to understand.  For example:

“Controls the development of the male specific abdominal muscle of Lawrence.”

The discussion of fruitless in the introduction is confusing.

The author might consider trying the free version of the online tool Grammarly to assist with continued proofreading of their manuscript.

For Fig 1, Fig 2, Fig 3, the axes in the version sent for review are not readable. Please be sure the versions in the final manuscript are easily readable. Ideally, in vector format.

For the microinjection method, there should be more details. E.g., what type of needle was used, if a commercial part # was used, that should be listed.

The reviewer was not able to find the newly Genbank accessions in Table 1. So the editor should confirm that the presented Genbank sequences are set to release upon publication.

Overall this manuscript is carefully done, and admittedly incremental, but also exciting.  I recommend this manuscript be published, after fixing of the grammatical issues, and the relatively minor issues above.

Reviewer 2 Report

This manuscript deals with the reference genes for RT-qPCT analysis in a single firefly species. Although this manuscript is potentially interesting and could be published, there are certainly some parts which should be considerably improved. I put all my comments and suggestions directly into the PDF file attached. I also have to mention that I am not very familiar with many methods and programs used in this study so I hope that another selected reviewer(s) can review that specific part of this manuscript.

My main concerns are as follows:

1) English needs to be substantially improved, as well as flow of the Introduction which is rather chaotic and very hard to read and understand...

2) Conclusions given in Abstract do not correspond at all with the results described in Results in Table 2 (different genes)...

3) There are quite a number of inconsistencies and errors throughout the text which should be improved (see the PDF attached).

4) In figure of an adult beetle, there is a structure called "larval organ" which is not discussed in the manuscript and I ahve never heard about it. This should be explained. Who did you follow in using the morphological terminology?

I highly recommend some improvements are made in the manuscript before it can be accepted for publication.

Reviewer 3 Report

Dear author, 

I believe this research has scientific merit, as it is an unprecedented study that will be a reference for future molecular studies in Lampyridae. Nevertheless, the English language needs a wide revision. I marked in the pdf some unclear phrases and terms, but the whole text needs revision. 

Round 2

Reviewer 2 Report

Thank you very much for significant improvement of the manuscript. As it is, it is almost acceptable for publication, with only a few minor things which I list below:

line 14 - comma between "first, example" is not needed

line 15 - Aquatica can be abbreviated (A. leii)

line 92 - authors added "A. leii breeding lab" but this does not make sense... should there be "breeding in the lab"? or is there any "A. leii breeding lab" = laboratory for breeding specifically this species? Please explain or edit.

Figure A2 - "larval light organ"

  • this concept of LLO needs some further explanation in the manuscript so that readers can understand what is it and why it is called "larval" although it is on the adult female (some very brief intro of this problem and explanation that it is on different abdominal segment than adult light organ, etc. with some reference would be enough).

Once more many thanks for the improvements and congratulations to an interesting study.

Author Response

Further revisions are below:

  1. comma between "first, example" deleted.
  2. Aquatica leii in line 15 abbreviated as A. leii
  3. line 92, A. leii breeding lab was explained in detailed as "A. leii breeding lab (a lab established solely to breed solely A. leii, the original firefly population was collected from Hangzhou city) "
  4. Figure A2 - "larval light organ", detailed explanation added "LLO, larval light organ (larval firefly light organs function from larval stages to pupual stages and 24 hours after adult emergence)"

Reviewer 3 Report

Dear authors

I believe the manuscript has been significantly improved and now it has merit to be published in Isects, with only minor corrections:

Line 14 – after "first" delete virgule.

Line 2630 – Coleopetera => Coleoptera

Author Response

Revsions made by suggestions of reviewer:

1. Line 14 – after "first" virgule deleted

2. Line 2630  Coleopetera chagned to Coleoptera